# Many-body localization in a quantum Ising model with the long-range interaction: Accurate determination of the transition point

Illia Lukin,[1] Andrii Sotnikov,[1, 2] and Alexander L. Burin[3]

[1]*National Science Center "Kharkiv Institute of Physics and Technology", Akademichna str. 1, 61108 Kharkiv, Ukraine*
[2]*V.N. Karazin Kharkiv National University, Svobody Square 4, 61022 Kharkiv, Ukraine*
[3]*Department of Chemistry, Tulane University, New Orleans, LA 70118, USA*
(Dated: May 30, 2025)

Many-body localization (MBL) transition emerges at strong disorder in interacting systems, separating chaotic and reversible dynamics. Although the existence of MBL transition within the macroscopic limit in spin chains with a short-range interaction was proved rigorously, the transition point is not found yet because of the dramatic sensitivity of the transition point to the chain length at computationally accessible lengths, possible due to local fluctuations destroying localization. Here we investigate MBL transition in the quantum Ising model (Ising model in a transverse field) with the long-range interaction suppressing the fluctuations similarly to that for the second-order phase transitions. We estimate the MBL threshold within the logarithmic accuracy using exact results for a somewhat similar localization problem on a Bethe lattice problem and show that our expectations are fully consistent with the estimate of the transition point using exact diagonalization. In spite of unlimited growing of the critical disorder within the thermodynamic limit, this result offers the opportunity to probe the critical behavior of the system near the transition point. Moreover, the model is relevant for the wide variety of physical systems with the long-range dipole-dipole, elastic or indirect exchange interactions.

PACS numbers: 73.23.-b 72.70.+m 71.55.Jv 73.61.Jc

**Introduction**. In contrast to a single-particle localization [1] emerging if the interaction is negligible that usually takes place in the zero temperature limit, Many-body localization (MBL) transition can take place at arbitrary temperature. MBL transition separates chaotic (delocalized) and deterministic (localized) dynamic phases in interacting quantum systems. Chaotic phase satisfies eigenstate thermalization hypothesis [2, 3] suggesting that the system serves as a thermal bath for any part of it, establishing thermal equilibrium in each part, while the system does not get to the equilibrium in the localized phase. Transition to equilibrium in a chaotic regime is accompanied by energy equipartition, i.e., irreversible relaxation erasing memory about the initial state of the system, while the localized system remembers its initial state forever. Consequently, the localized phase is desirable in modern quantum devices including quantum hardware [4–9], where the relaxation inevitably destroys the memory.

In spite of numerous efforts targeted to understand and characterize MBL transition in various systems [10–21], it is not clear yet: (A) How to determine the MBL transition point? and (B) How does the system behave in the critical domain near the transition? The localization threshold can be relatively easy determined numerically in a single-particle problem, where the system complexity (size of the eigenstate basis) grows proportionally to the power of the system size [22], while in the MBL problem that complexity grows exponentially [15, 23] and does not converge well with increasing the system size for numerically accessible sizes of the spin chain with short-range interactions, where the existence of MBL transition in the thermodynamic limit of the infinite size has

been rigorously proved [21, 24]. *The investigation of the MBL transition in the spin system with the long-range interaction reported in the present work is capable to address these challenges*, because the MBL transition can be determined there analytically exploiting its similarity with the solvable localization problem on the Bethe lattice [25], which is the main focus of the present work. The knowledge of the transition point should help to investigate the system behavior within the critical domain.

Generally, MBL problem can be formulated using the Hamiltonian split into static and dynamic parts as [13, 14]

$$\widehat{H} = \widehat{H}_0 + \xi \widehat{V} \tag{1}$$

where $\widehat{H}_0$ is a static Hamiltonian composed of commuting operators $\widehat{A}_i$ (e.g., spin projection operators $S_i^z$ to the $z$ axis defined in local sites $i$) and $\widehat{V}$ is a dynamic interaction including operators not commuting with each other and the static Hamiltonian (e.g., spin projection operators $S_i^x$ to the $x$ axis, as in Eq. (2) below). The parameter $\xi$ (transverse field $\Gamma$ in Eq. (2)) controls the strength of dynamic interaction. The system is obviously localized for $\xi = 0$, and the physical parameters characterized by the operators $\widehat{A}_i$ serve as its integrals of motion. The static eigenstate basis $\{B\}$ of the system Fock space is chosen using the states with given values of operators $\widehat{A}_i$ (for $N$ interacting spin-1/2 particles this will be $2^N$ states, characterized by different spin projections $S_i^z = \pm 1/2$ to the $z$ axis). The Fock space for a single-particle localization problem involves only $N$ states defined, for example, as a single spin excitation of the ferromagnet state with the total spin projection to the $z$ axis

$\pm(N/2 - 1)$, provided that the dynamic interaction $\widehat{V}$ does not modify the spin projection.

If the dynamic parameter $\xi$ differs from zero, then dynamic interactions modify eigenstates that are represented by a superposition of states belonging to the static basis with fixed operators $\widehat{A}_i$. For a single-particle problem, this modification can be expressed in terms of the number of states from the static basis set involved into a single eigenstate. If this number does not change with the system size, the states are localized, while if it increases, then they are delocalized. Chaotic regime is characterized by the Wigner-Dyson level statistics [26] emerging due to level repulsion. Delocalized states (except for the marginal case of the transition point) occupy the whole space, and there is substantial level repulsion between them lacking for localized states, located far away from each other, that leads to the Poisson level statistics.

In the typical MBL problem, the number of states from the static basis set involved into a single eigenstate grows with the system size even in the many-body localization regime [27], but slower than the size of the basis set. Emergence of the Wigner-Dyson statistics serves as an indication of a chaotic regime [20, 27], where all states are communicating with each other leading to substantial level repulsion. There are other criteria for the MBL transition using local correlation function in the infinite time limit [28, 29], local integrals of motion [30] or time evolution of entanglement entropy [31]. The consideration of level statistics is easiest computationally, because it uses only energies of selected eigenstates, which can be found using sparse matrix diagonalization algorithms applicable to relatively large systems.

It was proved rigorously [24] that the MBL phase emerges in certain spin chains with short-range interactions at sufficiently small (yet nonzero) dynamic interactions (parameter $\xi \neq 0$). However, advanced numerical investigations of the localization threshold in such chains with the number of spins as large as $N = 25$ [32–34] converge very slowly, possibly due to a system sensitivity to local fluctuations [35], often leading to a localization breakdown. The conclusive results can be attained investigating a system of more than 30 spins, which is not accessible by the classical computer. There is no analytical approach to the localization threshold in the MBL problem with a short-range interaction, similarly to the Anderson localization problem.

Yet there is one exception—the localization on a Bethe lattice, where the localization threshold is derived analytically [36] and it is consistent with advanced numerical studies [37], see also Refs. [38, 39]. Here we examine the MBL problem for the system of $N$ interacting spins coupled with each other by random static binary interaction independent of interspin distance [40]. We argue that the localization threshold within this model is approximately three times smaller than that in the Bethe lattice and estimate it within the logarithmic accuracy following the arguments of Ref. [40]. Then we estimate the position of the localization threshold using level statistics [20]

and show that it is perfectly consistent with the analytical predictions. Knowledge of the localization threshold opens up the opportunity to directly examine the system behavior within the critical domain of the transition, which is postponed for future work.

One additional comment is in order. The long-range interaction is almost ubiquitous in any system of interacting quantum objects (spins, electrons or atomic tunneling systems) due to dipole-dipole, elastic, magnetic dipole or indirect exchange interactions [29, 41, 42]. Therefore, the present work is potentially applicable to many physical systems, where many-body localization has been observed, including interacting two-level systems in amorphous solids [41, 43, 44], nitrogen vacancies in diamond [45], and cold ions [46]. Certainly, these systems were analyzed theoretically and reported in numerous publications [14, 29, 40–42, 47–51], and there exists common understanding of the nature of MBL transition based on the criterion of a single resonance per state similar to that for the Anderson localization problem in three dimensions. Within the present study, we attempt to characterize MBL transition not only at the qualitative, but quantitative level estimating accurately the MBL threshold.

**Model and Bethe lattice counterpart.** We investigate the quantum spin glass model of $N$ interacting spins characterized by the Hamiltonian having the form similar to Eq. (1) (see Refs. [40, 52])

$$H = \frac{1}{2} \sum_{i \neq j} J_{ij} S_i^z S_j^z + 2\Gamma \sum_i S_i^x, \qquad (2)$$

where $S_i^\mu$ are spin $1/2$ operators ($\mu = x$, $y$ or $z$). The first term is the static Hamiltonian representing the celebrated Sherrington-Kirkpatrick model [53] with random, uncorrelated interactions $J_{ij}$ characterized by the Gaussian distribution with $\langle J_{ij} \rangle = 0$ and $\langle J_{ij} J_{kl} \rangle = J_0^2 \left( \delta_{ik}\delta_{jl} + \delta_{il}\delta_{jk} \right)$, while the second term corresponds to the transverse field. In the absence of the dynamic term ($\Gamma = 0$), eigenstates of the model are represented by the $2^N$ vertices of $N$-dimensional hypercube with each vertex coordinates given by $N$ spin projections $S^z = \pm 1/2$ for the corresponding state, as illustrated in Fig. 1(a) for three spins. Previously, the MBL problem has been investigated for a random energy model [52, 54]. The phase space of this model can similarly represented by a hypercube. Yet, the quantum random energy model is equivalent to the Anderson localization problem on the hypercube, while the present model possesses a strong correlation between different state energies [40] and destructive interference similarly to the MBL problems with short-range interactions. We consider the present model as a bridge between single-particle and many-body localization problems.

We are interested in the MBL transition at the infinite temperature for the states with total energy close to zero. Then the energy difference between adjacent hypercube vertices $m$ and $n$ (see Fig. 1(a)) is given by a spin-flip energy $\omega_i = 2S_i^z \sum_j J_{ij} S_j^z$.

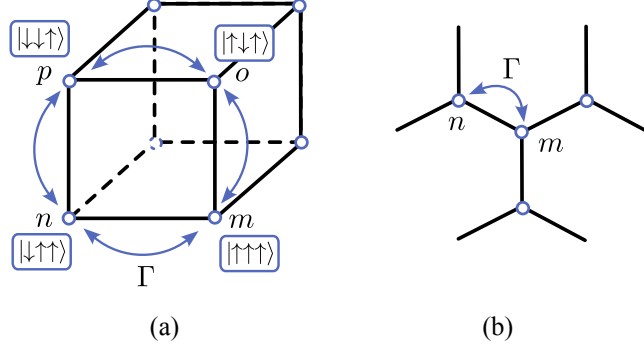

FIG. 1: (a) Hypecube with vertices representing the phase space of three spins. (b) Bethe lattice with each site coupled to three neighbors.

The matching Bethe lattice problem is constructed following the recipe of Refs. [15, 40, 55]. Each state of the spin system forms the vertex in the graph coupled with $N$ other states different from this one by a single spin flip and the coupling strength is given by the dynamic parameter $\Gamma$. Thus, the number of neighbors in the matching Bethe lattice model is $N$. The effective distribution of random potentials is given by that of the spin-flip energies $\omega_i$ [40], which corresponds to the Gaussian distribution defined as

$$P(\omega) = \frac{1}{\sqrt{2\pi}W} e^{-\frac{\omega^2}{2W^2}}, \quad W = \sqrt{N-1}\frac{J_0}{2}. \qquad (3)$$

The localization threshold in the Bethe lattice problem described above in the limit $N \gg 1$ is determined in Ref. [25] (see Eq. 7.7 there), which can be approximated by (we used the present notations and $\Gamma_c$ stands for the localization threshold)

$$\begin{aligned} 1 &\approx 2(N-1)\Gamma_B \int_0^\infty (P'(\omega) - P'(-\omega)) \ln\left(\frac{\omega}{\Gamma}\right) d\omega \\ &= 4(N-1)\frac{\Gamma_B}{\sqrt{2\pi}W} \ln\left(\frac{1.06W}{\Gamma_B}\right). \end{aligned} \qquad (4)$$

Equation (4) determines the localization threshold of the Bethe lattice problem, and we use it below to evaluate the localization threshold for the matching MBL problem in the interacting spin system characterized by the Hamiltonian (2).

**Analytical estimate of the MBL threshold.** The phase space is organized differently for the quantum Ising model shown in Fig. 1(a) and the corresponding Bethe lattice problem shown Fig. 1(b). In the former case, the graph representing the system (hypercube) has plenty of loops, while there are no loops in the latter case. Loops result in destructive interference for the hypercube, which is absent in the Bethe lattice [40].

The destructive interference is realized already in the second-order process involving two spins flips illustrated in Fig. 1(a) as the transition between the initial state $m$ and the final state $p$ that can happen by means of the flip of spins $i$ and $j$. If the spin $i$ transfers first, then

the transition goes through the vertex $n$ as $m \to n \to p$, while in the opposite case it goes through the vertex $o$ as $m \to o \to p$. The total transition amplitude in the second order of perturbation theory in the dynamic field $\Gamma$ is given by the sum of contributions of two paths, which is defined as $\Gamma_{ij} = \Gamma^2(\omega_i^{-1} + \omega_j^{-1})$ (remember that notation $\omega_k$ refers to the flip energy of the spin $k$).

The second-order process is significant in the resonant regime, where the total energy change due to two spin flips $\omega_{ij} = \omega_i + \omega_j - J_{ij}^*$ ($J_{ij}^* = 4J_{ij}S_i^z S_j^z$ and spin projections $S_i^z$, $S_j^z$ correspond to the static basis state $m$) approaches zero. In this resonant regime, the second-order transition amplitude reads

$$\Gamma_{ij} = \frac{\Gamma^2 J_{ij}^*}{\omega_i \omega_j}. \qquad (5)$$

The second-order transition amplitude approaches zero in the absence of interaction between spins $i$ and $j$, reflecting the many-body nature of the two spin flip contribution lacking if the spins are just in static longitudinal fields, where no MBL transition can, obviously, take place.

The second-order amplitudes are substantially suppressed, if the spin-flip energies are large compared to the interaction between spins, i.e., $|\omega_i| \approx |\omega_j| \gg |J_{ij}^*|$. In the opposite cases of either $|\omega_i| \ll |J_{ij}^*|$ and $|\omega_j| \approx |J_{ij}^*|$ or $|\omega_j| \ll |J_{ij}^*|$ and $|\omega_j| \approx |J_{ij}^*|$, the contributions of two paths do not interfere with each other and contribute as two independent paths similarly to that in the Bethe lattice. This suggests the constraint of the maximum spin-flip energy in the definition of the localization threshold given by Eq. (4) by the spin-spin interaction $J_0/2$. Setting this constraint in the integral in Eq. (4) we obtain the equation for localization threshold within the logarithmic accuracy in the form

$$\beta \approx 4(N-1)\frac{\Gamma_c}{\sqrt{2\pi}W} \ln\left(\frac{\eta J_0}{\Gamma_c}\right), \qquad (6)$$

where $\beta$ and $\eta$ are unknown parameters of the order of unity and it is expected that $\beta > 1$ [40]. The parameter $\beta > 1$ originates from the higher-order processes determining the localization threshold. It is equal to unity for the localization problem on the Bethe lattice, but here it can be different because of the difference in the total number of distinguishable system pathways. Indeed, the total number of $n$-step paths in the Bethe lattice from the given point scales as $N(N-1)^{n-1}$, while for the interacting spin problem it scales as $N!/(N-n)!$ for the number of pathways [40]. Consequently, if the MBL transition is determined by the higher-order processes of the order of $N$, then the critical dynamic field $\Gamma_c$ can increase due to the reduction in the number of pathways, as reflected by the additional parameter $\beta > 1$ in Eq. (6).

Our derivation assumes that $\Gamma < J_0$, which is obviously true near the MBL threshold, since the width $W$ scales with the number of spins as $J_0\sqrt{N}$, thus we get $\Gamma_c^{\text{MBL}} \sim J_0/(\sqrt{N}\ln(N))$ similarly to the Bethe lattice.

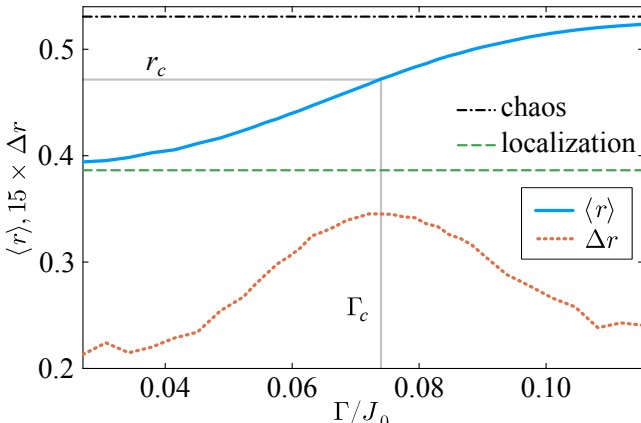

FIG. 2: General behavior of $\langle r \rangle$ and $\Delta r$ depending on the dynamic field $\Gamma/J_0$. The data is shown for $N = 13$.

Consequently, $\Gamma_c \ll J_0$ in the regime of interest $N \gg 1$. Below we report the investigation of MBL transition using the exact diagonalization of the Hamiltonian (2). **Exact diagonalization results for the MBL threshold.** To determine quantitatively the critical field strength $\Gamma_c$, we analyze the level statistics expressed through the gap ratio parameter $r$ [20]

$$r = \frac{\min\left(\Delta_n, \Delta_{n+1}\right)}{\max\left(\Delta_n, \Delta_{n+1}\right)}, \tag{7}$$

where the quantities $\Delta_n = E_{n+1} - E_n$ represent the differences in energies $E_n$ of the adjacent eigenstates (energy gaps). This ratio is then averaged over some energy window. The localized phase is characterized by the average gap ratio $r \approx 0.3863$, while in the delocalized (chaotic) phase $r \approx 0.5307$, where the average of the gap ratio is computed either over the full energy spectrum or over the specified energy window for a fixed choice of disorder.

One of the key observables in the current study is the variance $(\Delta r)^2 = \langle r^2 \rangle - \langle r \rangle^2$, where the averages are taken over disorder samples. This quantity was previously studied in Ref. [56] in a broader context of the gap ratio distributions between different disorder samples. There, it was observed that the variance has a maximum near the MBL transition point. Hence, it is natural to expect that the maximum fluctuation of the gap ratio emerges at the transition point. Indeed, the transition point is usually estimated as the inflection point in critical parameter $\langle r \rangle$ dependence on the disorder strength (e.g., the ratio $R = \Gamma/J_0$). If we assume that the ratio $r$ is a function of that parameter, one can represent the fluctuations of $r$ in terms of the fluctuations of $R$ as $\langle \delta r^2 \rangle \approx (dr/dR)^2 \langle \delta R^2 \rangle$. Since the fluctuation in disorder strength is not strongly sensitive to the transition point, the maximum should be observed at the maximum derivative $dr/dR$ corresponding to the inflection point.

This expectation is consistent with our studies showing that the maximum fluctuation of the average ratio $r$ emerges close to the transition region between the chaotic and localized behavior. This is illustrated in Fig. 2. We suggest using the point of maximum fluctuation $\Delta r$ to determine the phase transition point $\Gamma_c(N)$. Note that traditionally the MBL transition point is determined with the finite-size scaling analysis, where the data for $\langle r \rangle$ with different system sizes $N$ are collapsed on one curve with a certain scaling ansatz [57, 58]. This methodology relies on the assumption that for large system sizes, the transition point converges to a certain finite value. Although this may be the case for certain MBL systems with short-range and long-range interactions [57–59], the current analysis of the MBL transition in the Sherrington-Kirkpatrick model points towards the $N$-dependent transition point. Hence, we need a reliable criterion for the transition for the fixed value of $N$.

From the computational perspective, it is unfeasible to reach large system sizes ($N > 14$) using the full exact diagonalization of the Hamiltonian matrix (2) partially due to a rather dense structure of the latter (long-range interactions and almost no symmetries) at moderate $N$ and the exponential growth of the matrix size with the further increase of $N$. This brings us to the necessity to employ the filtering approaches, which allow us to diagonalize the Hamiltonian only in the chosen energy window of the full Hilbert space.

The idea of the filtering follows closely the celebrated Lanczos algorithm to determine the largest (smallest) energy eigenvalues and eigenvectors of the Hamiltonian. The main difference is that instead of the original Hamiltonian $H$, one needs to use a localized function $f(H)$ of the Hamiltonian. This function must be sufficiently small outside of a fixed energy window $E \in [E_{min}, E_{max}]$. There are several possible approaches on how this function can be defined and calculated [60–64]. In this study, we closely follow Ref. [33] (see also Ref. [65] for applications to Floquet systems), which proposes the polynomially filtered exact diagonalization method. This method constructs the function $f(H)$ as a finite-order Chebyshev polynomial expansion of the delta function. The number of polynomials in the expansion controls the size of the energy window. Typically, it appears sufficient to choose the energy window covering approximately 500 eigenstates of the Hamiltonian, precisely in the center of the energy spectrum of the model. Note that the model Hamiltonian has an additional $\mathbb{Z}_2$ symmetry upon the simultaneous flip of all spins. This symmetry is used to project the Hamiltonian onto the subspace with an additional eigenvalue of this symmetry generator. This projection reduces the effective Hilbert space dimension twice, and thus improves the speed and memory consumption.

Our numerical analysis is performed as follows: At the given system size $N \in [12, 18]$, we diagonalize the model (2) in a certain energy window for $\mathcal{N}_{dis}$ samples of the random couplings $J_{ij}$ with the fixed amplitudes $J_0$ and $\Gamma$. We take the number of disorder samples as $\mathcal{N}_{dis} = \{20000, 31200, 31400, 5400, 800, 400, 400\}$ for $N = \{12, 13, 14, 15, 16, 17, 18\}$, respectively. For every

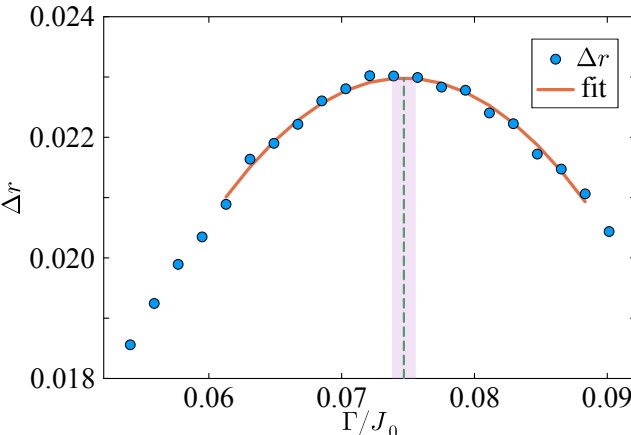

FIG. 3: Dependence of $\Delta r$ on $\Gamma/J_0$ for $N = 13$ and $\mathcal{N}_{dis} = 31200$ (circles) with the corresponding parabolic fit (solid line). The shaded area corresponds to the estimate of the error in determining position of $\Gamma_c$.

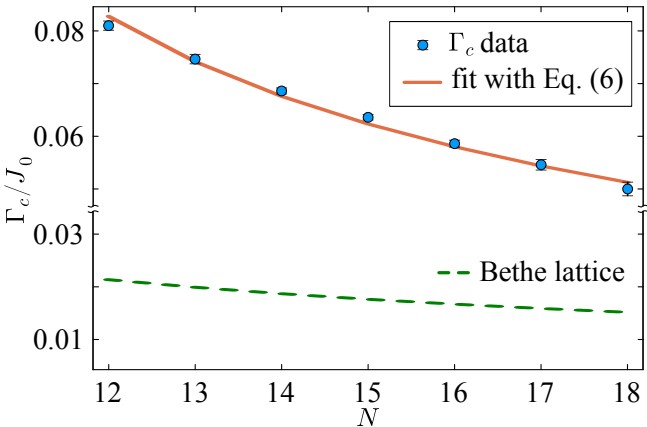

FIG. 4: Dependence of the transition point $\Gamma_c/J_0$ on the system size. The fitting coefficients are $\beta = 1.4$ and $\eta = 0.41$.

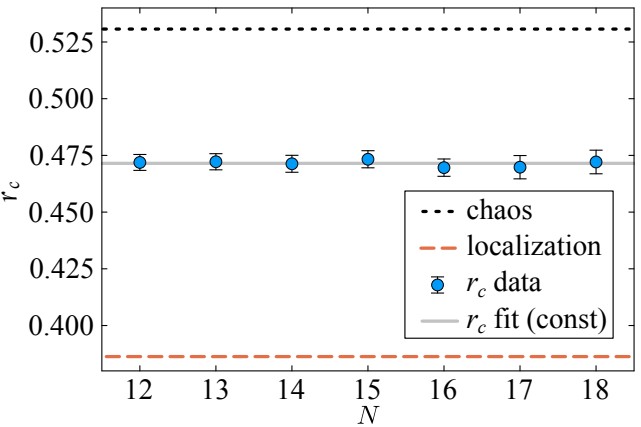

FIG. 5: Dependence of the value of mean gap ratio at the MBL transition point $r_c$ on the system size. The best horizontal fit corresponds to approximately $r_c^{(\mathrm{fit})} \approx 0.472$.

choice of couplings, we compute the value of the gap ratio (7) averaged over the predefined energy window. Next, both the mean $\langle r \rangle$ and deviation $\Delta r$ of this gap ratio are calculated by sampling over different disorder realizations. The resulting quantities are functions of both $N$ and $\Gamma/J_0$. The variance $\Delta r$ as a function of $\Gamma/J_0$ has a clear maximum for every value of $N$. As mentioned above, we take this maximum position as a transition point between chaotic and localized behavior for the given system size $N$. Note that since $\Delta r$ is estimated statistically, it still contains noisy features, which can affect the accuracy of the determination of $\Gamma_c$. To minimize the possible influence of these errors, we perform an additional fit of the numerical data $\Delta r(\Gamma/J_0)$ in the vicinity of the maximum with the parabolic function (both data and parabolic fit are illustrated in Fig. 3). To estimate the error bars of each point $\Gamma_c(N)$, we vary the size and position of the window of the fit, and also perform fits with twice smaller numbers of disorder realizations. The spread between the obtained values of the transition point defines the corresponding error bars.

Our main goal is to study the dependence of the MBL transition threshold on the system size. We adapt the ansatz in accordance with Eq. (6), where both $\beta$ and $\eta$ are determined numerically from the least-square fitting of the data points $\Gamma_c(N)$. The best fit predictions are $\beta \approx 1.4$ and $\eta \approx 0.41$. Both numerical results and predictions from the fit are shown in Fig. 4. In this figure, we also illustrate the results obtained for the Bethe lattice with Eq. (4). It is clear that the localized phase is more stable against the external field $\Gamma$ than in the Bethe lattice model.

Finally, we also investigate the correlation between the maximum position of $\Delta r$ and the value of $\langle r \rangle$ at the same critical field $\Gamma_c/J_0$, denoted by $r_c$ (see also Fig. 2). The dependence of $r_c$ on $N$ is shown in Fig. 5, where it exhibits nearly constant behavior in the studied range of system sizes.

**Summary.** We investigated finite-size scaling of Many-Body localization transition in the quantum Ising model and compared the results with matching Bethe lattice problem. In agreement with analytical expectations, the critical dynamic field $\Gamma_c$ for the MBL transition exceeds that for the Bethe lattice ($\Gamma_B$) by a numerical factor of order of unity, i.e., $\Gamma_c \approx 3\Gamma_B$. Analytical equation for the MBL transition, Eq. (6), describes very well the dependence of the critical dynamic field on the number of spins, as confirmed by the exact numerical diagonalization, if we set $\beta = 1.4$ and $\eta = 0.41$. The weak dependence of the critical value of the average ratio $r$ at the transition point shown in Fig. 5 suggests that the finite-size effects are not significant already for $N = 18$ spins, thus the reported estimates of the transition point are rather accurate. This is in a striking contrast with the MBL transition in the system with the short-range in-

teraction [32], where one needs to approach much larger sizes to characterize MBL transition.

Therefore, according to our analytical and numerical studies, the MBL transition can be identified reasonably well for the quantum Ising model with the infinite range of interaction using the analytical results for the Bethe lattice problem. This model can be used for the characterization of critical system behaviors near the MBL transition point.

## ACKNOWLEDGMENTS

This work is supported by the Tulane University Lavin Bernick Fund. I.L. acknowledges support by the IMPRESS-U grant from the US National Academy of Sciences via STCU project No. 7120 and the IEEE program Magnetism for Ukraine 2025, Grant No. 9918. A.S. acknowledges support by the National Research Foundation of Ukraine, project No. 0124U004372. A.B. also acknowledges the support by the National Science Foundation (CHE-1462075).

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

tion in Floquet systems," Phys. Rev. B **107**, 115132 (2023).