# Peer review of "Many-body localization in a quantum Ising model with the long-range interaction: Accurate determination of the transition point"

_SciPost Physics_

## Round 4 · Referee Report · Anonymous (Referee 1) · 2025-12-24

Report

The authors' changes are sufficient.

Recommendation

Publish (easily meets expectations and criteria for this Journal; among top 50%)

---

## Round 4 · Author Response

Dear Sir or Madam,

Thank you for considering our manuscript. We greatly appreciate the remarks and suggestions made by both referees and we did our best to address them all in the revised version that we are resubmitting for your consideration.

Sincerely yours,
Illia Lukin, Andrii Sotnikov and Alexander Burin

Below are our responses to the Referee Comments and description of changes. We also have the revised version where the changes are shown in colors. It can be submitted upon request.

Referee 1 Comment 1: The last and one before last paragraphs on p3 appear to be duplicated

Author response: Thank you, corrected.

Referee 1 Comment 2: The authors use a non-standard form of the SK model where the random couplings are not normalized, so that in the thermodynamic limit, the disordered term of the Hamiltonian is super-extensive (scales like N^2). The authors should at least bring attention to or explain this specific choice.

Author response: Thank you for pointing this out. Our original choice was motivated by the desire to keep all Hamiltonian parameters independent of the number of spins, which appears to us the most natural convention. The normalization by the square root of the number of spins is required to ensure that the spin-glass transition temperature remains independent of system size. However, this normalization does not make the localization threshold at infinite temperature size-independent in the thermodynamic limit: the critical transverse field vanishes as the number of spins tends to infinity.

Following the referee’s recommendation, we have added a discussion of the normalization choice to Sec. 2. In addition, we introduce an alternative normalization that yields a convergent localization threshold in the thermodynamic limit; this is now presented at the end of Sec. 4 (second paragraph after Eq. (9)).

Referee 1 Comment 3: In Eq(4) P' is probably P. If not, it should be defined.

Author response: Thank you for pointing out this issue. Indeed, it is the derivative of the probability density P. We have clarified its definition in the revised manuscript Appendix, following carefully the original work of Abou-Chacra and coworkers.

Referee 1 Comment 4: For reproducibility, the calculation of the second line in Eq. (4) should be detailed, maybe in the appendix, since Ref. 25 gives a calculation for a uniform distribution.

Author response: The requested details have been added to the Appendix in the revised manuscript, as recommended by the Referee.

Referee 1 Comment 5: Figure 2 is confusing due to the combined y-axis. I suggest splitting into two panels. This maintains the point of the figure while adding clarity.

Author response: Figure 2 is split as recommended by the Referee.

Referee 1 Comment 6: The authors obtain a power-law decay in system size critical coupling strength, which is consistent with my comment on the dominance of the disordered term in the thermodynamic limit. What would be the scaling of the critical coupling strength for the standard SK normalization of the disordered strength? This point should be discussed in the conclusion/discussion.

Author response: The discussion is added to the conclusion as recommended by the Referee.

Referee 2 question 1: What is the behavior of other characteristic points of the ergodic–MBL crossover in the studied model? For example, does the Bethe-lattice prediction also describe (i) the crossing point of the r curves, or (ii) the onset of the deviation of r from its ergodic value?

Author response: Following the referee’s suggestion, we have attempted to represent the transition points as the crossing points of the r curves, in line with earlier work by one of us and his graduate student. We find that a reasonable fit is obtained using our previously determined critical transverse fields, under the assumption that the transition width decreases with the number of spins N as N to the power -3/2. This analysis has been incorporated into Sec. 4 of the revised manuscript, and the behavior of the transition width is also discussed in the Conclusion.

Referee 2 question: The quantum random energy model (studied recently also in Phys. Rev. B 111, 214206 (2025) is more directly amenable to Bethe-lattice arguments because energies on neighboring hypercube vertices are uncorrelated. In contrast, the present Ising model exhibits correlations. While the presence of the correlations make the presently studied model more interesting from the many-body perspective, do the authors have any argument why the Bethe lattice argument is applicable despite the correlations?

Author response: This is indeed an important remark. The energies in the system are correlated, so that the energy change due to a single spin flip is much smaller than the typical energy of a Sherrington-Kirkpatrick model state. However, eigenstates with energy E close to zero are formed in both problems by sequences of resonant sites having approximately the same energy (e.g.,
E=0). For the spin-glass problem, the energies of neighboring sites differ by spin-flip energies that approach zero for sites with identical energies. Consequently, the identity of energies along a delocalization path is equivalent to having nearly zero spin-flip energies along this path.

Since spin-flip energies for different spins have negligible correlations at large
N and at infinite temperature, these correlations can be safely neglected, as discussed in earlier work by one of the authors (Ann. Phys., NY 529, 1600292; see also the preprint https://arxiv.org/abs/1610.00811). We refer explicitly to the preprint because it includes a Supporting Information section with detailed and accurate derivations that were omitted in the published paper. The author (AB) has contacted the Annals of Physics editorial office requesting the missing Supporting Information be added to the online version, and they have kindly agreed. We hope that this updated version will be available on the journal website at the time of manuscript consideration.

The discussion of the absence of correlations between spin-flip energies has been added to the revised manuscript (Sec. 2, paragraphs 2 and 3), along with a comparison of our results to those of the quantum random energy model (Sec. 4, paragraph following Eq. (9)). In addition, the Introduction and Conclusion have been revised to emphasize the similarity of the present problem to the Bethe-lattice problem.

Referee 2 question 3: For the analytical estimate of the MBL threshold, the authors consider only second-order processes. However, higher-order processes have been argued to play a significant role in MBL physics (see, e.g., arXiv:2005.13558; Phys. Rev. B 104, 184203 (2021); SciPost Phys. 12, 201 (2022); Phys. Rev. Lett. 131, 106301 (2023)). Do the authors have arguments for why such processes do not invalidate the Bethe-lattice threshold in the present model?

Author response: This is indeed an important question, as it highlights the difference between the present model and systems with short-range interactions, including those cited by the referee. In our model, the breakdown of localization is reasonably well determined—within logarithmic accuracy—by the condition of one resonance per eigenstate of the static Hamiltonian
H0 (Eq. (1)). This is analogous to the definition of the Anderson localization threshold in three dimensions or on the Bethe lattice, and it is fundamentally different from the MBL transition in paradigmatic short-range spin chains.

In short-range spin chains, the single-resonance criterion yields a localization threshold for the dynamic interaction that vanishes in the thermodynamic limit as 1/N, which is not the case. Considering second-order processes in short-range models produces vanishing two-spin hopping amplitudes (Eq. (5)) for the vast majority of spin pairs, since most of them are non-interacting. This dramatically alters the single-spin-flip estimate of the localization threshold, in stark contrast to our model, where all spins interact with each other. A note on this distinction has been added to the Introduction and at the end of Sec. 3.

As discussed in earlier work (Ann. Phys., NY 529, 1600292; see also https://arxiv.org/abs/1610.00811), the inclusion of higher-order (n>2) spin transitions in the present model effectively rescales the argument of the logarithm in the localization threshold by the square root of the number of spin flips. Together with the reduced number of non-self-intersecting paths compared to the Bethe lattice, this is expected to modify the localization threshold only by a numerical factor relative to the Bethe-lattice result. This expectation is confirmed by the numerical results presented in the current work.

A discussion of higher-order processes has been added to Sec. 3 (just before the last paragraph).

Referee 2 question 4: Could the authors comment more extensively on how the variance Δr^2 is calculated? Is each ri taken into account as corresponding to a single disorder sample? Is this variance dependent on the number of energy levels within a single sample?”

Author response: We thank the Referee for the questions. These points were not explained clearly in the previous version. Note that, first, we average the gap ratio over a certain energy window to obtain its value at a fixed disorder realization. Next, we compute the variance of the obtained gap ratios over disorder realizations. The chosen size of the energy window can slightly alter the resulting value of the variance, because small window sizes typically introduce additional source of fluctuation. We added the corresponding text with clarification to the paper.

---

## Round 4 · List of Changes

Summary of Revisions

Section 1 - Introduction

Two explanatory paragraphs were added on page 3 (fourth and third from the end) to emphasize the similarity of localization transitions in the present problem and the single-particle localization problem.

A brief outline of the manuscript was added at the end of the section.

Section 2 - Model and Bethe Lattice Counterpart

A discussion of the normalization of interactions was added (page 4, second paragraph).

Considerations of the spin-flip energies, their probability density function, and their statistical independence were added (page 4, paragraph 3).

A detailed explanation of the Bethe lattice formulation of the problem was incorporated into paragraphs 5 and 6 (pages 4–5).

Section 3 — Analytical Estimate of the MBL Threshold

Two paragraphs were added at the end of the section (page 6), discussing the contributions of higher-order processes and the distinctions from the MBL transition in systems with short-range interactions.

Section 4 — Exact Diagonalization Results for the MBL Threshold

The paragraph justifying the use of level statistics was moved from the Introduction to the beginning of Section 4 (page 6).

A detailed explanation of how the gap-ratio data were collected was added (page 7, paragraph 3).

A comparison of the numerical results with those for the Bethe lattice problem was added at the end of page 9 and beginning of page 10.

The rescaling of the spin–spin interactions, which ensures that results remain independent of the system size, is now discussed in the second paragraph of page 10.

A comparison of the critical gap ratios with previously reported values for the quantum random energy model and for random regular graphs was added to the end of the third paragraph on page 10.

The description of the transition as the crossing point of the r-curves was added to the last paragraph of page 11. A new figure (Fig. 6) showing the rescaled data collapse was also included there.

Section 5 - Conclusion

A new first paragraph summarizing the main results, and a second paragraph comparing them with those of related models, were added on page 11.

Additional discussion of the methods was inserted as the second paragraph on page 12.

A discussion of the effect of rescaling was added at the end of the conclusion.

Appendix is added containing the estimate of the localization threshold for the Bethe lattice problem with the Gaussian distribution of site energies.

---

## Editorial Decision

in_refereeing